# Toxicity in Goats Exposed to Arsenic in the Region Lagunera, Northern Mexico

**DOI:** 10.3390/vetsci7020059

**Published:** 2020-05-04

**Authors:** Natalia B. Ortega-Morales, José A. Cueto-Wong, Eutiquio Barrientos-Juárez, Gonzalo García-Vargas, Homero Salinas-González, Alain Buendía Garcia, Javier Morán-Martínez

**Affiliations:** 1Facultad de Agronomía y Zootecnia, Universidad Juárez del Estado de Durango, Km. 35 Carretera Gómez Palacio-Tlahualilo, Domicilio conocido poblado ejidal de Venecia, Gómez Palacio C.P. 35111, Mexico; nabel_87@hotmail.com; 2Instituto Nacional de Investigaciones Forestales, Agrícolas y Pecuarias, Campo Experimental La Laguna, Blvd. José Santos Valdéz No. 1200 Pte., Matamoros C.P. 27440, Mexico; cueto.jose@inifap.gob.mx; 3Instituto Nacional de Investigaciones Forestales, Agrícolas y Pecuarias, Campo Experimental Aldama-La Campana, Km 33.5 Carretera Chihuahua-Ojinaga, Cd. Aldama C.P. 32910, Mexico; barrientos.eutiquio@gmail.com; 4Facultad de Ciencias de la Salud, Universidad Juárez del Estado de Durango, Calzada Palmas 1, Revolución, Gómez Palacio C.P. 35050, Mexico; ggarcia_vargas@hotmail.com; 5Cooperative Extension and Research, Lincoln University of Missouri, 900 Chestnut St., Jefferson City, MO 65101, USA; salinas-gonzalezh@lincolnu.edu; 6Facultad de Contaduría y Administración, Universidad Autónoma de Coahuila, Boulevard Revolución No. 151, Torreón Coahuila C.P. 27000, Mexico; 7Universidad Autónoma Agraria Antonio Narro, Unidad Laguna, Av s/n col. Valle Verde, Torreón, Coahuila C.P. 27053, Mexico; alain75191@hotmail.com; 8Departamento de Biología Celular y Ultraestructura, Centro de Investigación Biimédica, Facultad de Medicina, Universidad Autónoma de Coahuila Unidad Torreón, Gregorio A. García No. 198 sur. Torreón, Coahuila C.P. 27000, Mexico

**Keywords:** toxicity, goats, arsenic, water, urine

## Abstract

The Region Lagunera, a region in northeast Mexico, is undergoing significant problems with the quality of its groundwater, which exceeds the permissible limits of contaminants and/or heavy metals stipulated in Mexican legislation. The present study evaluated chronic toxicity in male goats exposed to arsenic via one ex situ Group 1 (n = 5) and one in situ female goats Group 3 (n = 10). The treatment in Group 1 was carried out in the La Laguna experimental field of the Instituto Nacional de Investigaciones Forestales, Agrícolas y Pecuarias (INIFAP), located in Matamoros, Coahuila, Mexico. Sodium arsenite (2 mg/kg) was orally administered for 84 days to five male Creole goats, aged between four and five years old and weighing between 60 and 70 kg, in order to determine its effect on urine toxicity, libido, and physiological condition, an untreated group (n = 5) was included (Group 2). The experiment in group 3 was conducted on ten female Creole goats, aged between four and six years old and weighing between 40 and 49 kg, in both the contaminated sampling area in the rural community of El Venado and the control sampling area in the rural community of Nuevo Reynosa (Group 4 (n = 5)), in which the arsenic levels were measured in the urine of the exposed goats, as was their physiological condition. Significant differences (*p* < 0.01) between the groups were found in both the arsenic concentration in the urine and the physiological condition observed in both experimental groups.

## 1. Introduction

The presence of arsenic in groundwater used for human or animal consumption is one of the most significant global health problems [1], causing chronic toxicity in exposed organisms [2]. The highest concentrations of arsenic in Mexico have been found mainly in the groundwater of the north and center of the country, with one of the most affected areas including large parts of the states of Coahuila and Durango, comprising the region known as the Region Lagunera, while other states, such as Chihuahua, Sonora and San Luis Potosí, have also been affected [3]. The presence of arsenic in the groundwater of the Region Lagunera is mainly a consequence agricultural overexploitation, with the concentrations of arsenic reported in the area varying from 7 to 740 μg L^−1^ [4]. The maximum permissible limits (MPL) established in Mexico for arsenic in drinking water under Official Standard NOM-127-SSA1-1994 stipulates an MPL of 50 μg L^−1^ [5], while the World Health Organization (WHO) stipulates 10 μg L^−1^ [6]. The MPL for animal consumption is 200 μg L^−1^ [7,8], while the MPL of arsenic in milk in Mexico is 0.2 mg kg^−1^, according to NOM-243-SSA1-2010 [9]. The presence of arsenic in groundwater is the main contaminant of the phreatic mantles, causing chronic toxicity to organisms exposed via consumption and, thus, posing a significant threat to human, animal and environmental health [10]. Arsenic toxicity in goats has been subject to prior research, mainly in India, Bangladesh and Pakistan, which has reported physiological effects after two weeks of arsenic exposure via drinking water [11]. Other effects of chronic arsenic exposure in goats are dehydration, loss of appetite, physical debilitation, hair loss, weakness, stress, severe abdominal pain, restlessness, audible discomfort, increased respiratory rate, salivation, teeth grinding, and diarrhea [11,12]. The Creole breed reared in the Region Lagunera presents characteristics pertaining to the Anglo-Nubian, French Alpine, Saanen and Toggenburg goat breeds, for which breeds no information is available on the toxic effects of chronic arsenic exposure. Therefore, this study aimed to evaluate chronic toxicity in goats exposed to high concentrations of arsenic via drinking water in the Region Lagunera.

## 2. Materials and Methods

### 2.1. Study Conditions

The Region Lagunera is located in the northeast of Mexico, comprising fifteen municipalities, of which ten correspond to the state of Durango and five to Coahuila, and a total area of 48,887.50 km^2^ [13]. With a semi-arid climate with strong seasonal variations and infrequent rainfall, water is mainly obtained in the region by aquifer drilling and extraction, with 91% of surface and groundwater used for agriculture, with the remainder pertaining to urban (5%) and livestock (2%) use, while the water used for industry (1%) is exclusively obtained from subterranean water sources [14]. A total of 25 Creole goats were recruited for the treatments and randomly allocated to each experimental group (four groups: Group 1 n = 5 (male goats); Group 2 n = 5 (male goats); Group 3 n = 10 (female goats) and Group 4 n = 5 (female goats)). The experimental subjects were handled under the guidelines set out in the Official Mexican Standard [15] regarding humanitarian treatment in the transport of animals. The present study also adhered to the guidelines set out in the Official Mexican Standard pertaining to the technical specifications for laboratory diagnostic tests conducted in the zoosanitary area [16] and the Official Mexican Standard pertaining to epidemiological surveillance [17]. In this study, the handling and care of the animals have been considered, as well as the ethical considerations corresponding to the institutional regulations (UAAAN/UL: 1330-8241-2868), as well as the national and international regulations: NAM, 2002 [18] and FAS, 2010 [19], respectively.

### 2.2. Identification of Arsenic in the Water at Locations in the Comarca Lagunera

To determine the levels of arsenic in the water, tests were carried out on water samples collected at different locations in the Region Lagunera between August and December 2016 (Table 1; Appendix A). Each 90 mL aliquot of water obtained from public water faucets was placed in a 100 mL Nasco^®^ brand hermetically sealed plastic bottle, with all samples appropriately labeled and transported to the Laboratory of Environmental Toxicology of the Faculty of Health Sciences at the Universidad Juárez del Estado de Durango (UJED), in the city of Gómez Palacio, Durango, Mexico. The samples were analyzed to determine pH and electrical conductivity, using a Combo-Hanna^®^ edge potentiometer, and then acidified with 2 mL of HNO_3_ and frozen at −70 °C for storage until subsequent analysis. Controls and calibration curves were created using 10% HCL blanks and deionized water. To determine the arsenic level, the samples were subject to a total acid digestion in 50 mL glass vessels, with 0.20 mL of H_2_SO_4_ and 1.0 mL of concentrated K_2_O_8_S_2_ added to 1 mL of water and placed on a heating plate at 80 °C for approximately 30 min or until total evaporation, and then adjusted to 9 mL with 10% HCl, to which 1 mL of 20% KI was then added. Determinations of total arsenic concentration were obtained using Excalibur Atomic fluorescence equipment (model 10.055 N/S 571 PSA, Analytical^®^).

### 2.3. Experimental Design

Two experiments, comprising an Ex situ (E.S.) and an In situ (I.S.) treatment, were conducted to determine the toxic effects of arsenic on goats.

#### 2.3.1. Group 1

Group 1 was considered as ex situ treatment. Group comprised five male Creole goats (crosses of Alpino, Saanen, and Nubia breeds), between the ages of four and five, of healthy appearance, weighing 60–70 kg each, and born in different communities of the municipality of Matamoros, Coahuila. The goats were transported to INIFAP’s La Laguna experimental field, also located in the municipality of Matamoros, Coahuila, Mexico, at 1100 m.a.s.l., which has a dry arid climate, average annual rainfall of 240 mm and an annual average temperature of 25 °C in the shade, with extremes of −1 °C in the winter and 44 °C in the summer [20]. The subjects were acclimatized in a new 5 × 6 m pen for 15 days, fed a diet consisting of a mixture of alfalfa, forage corn and 18% commercial crude protein concentrate, and given drinking water from one of the waterholes in the experimental field which had an arsenic concentration of 25 ± 10 μg L^−1^, a level considered low according to NOM-127-SSA1-1994. Each goat was given a daily oral dose of sodium arsenite (2 mg kg^−1^ PV) throughout the experiment. 

#### 2.3.2. Group 2

Additionally, another five goats from the same breeding sites were also transported to the experimental field, kept under the same conditions as the treatment group, and given drinking water from the same water source, but were not administered sodium arsenite. Group 2 was considered as control group.

#### 2.3.3. Group 3

Group 3 was considered as in situ treatment. The second treatment group consisted of ten female Creole goats (crosses of Alpino, Saanen, and Nubia breeds) of healthy appearance, with an average age of between four and five years, weighing 40–50 kg each. All animals in this group were kept in their original pens and were born in El Venado, a rural community in the municipality of Matamoros, Coahuila, with the highest concentration of arsenic and a high number of goats. The In situ treatment was carried out in an arsenic contaminated area of EL Venado, which has a semi-arid dry climate with, during the season sampled, an average temperature ranging from 18 to 22 °C. The goats belonged to a goat farmer, whose herd comprised approximately 600 mostly female goats fed by grazing on vegetation typical of the region, such as candelilla, agave, and other native flora. These subjects drank water from drinking fountains, which contained high concentrations of arsenic, and were kept in a special pen, from which they were let out to pasture from 6 am to 4 pm. The libido of the subjects in this group was not measured, while their physiological conditions, the effect of white muscle disease (Appendix A), and the arsenic levels in their milk were observed. The arsenic concentration in the water in El Venado was 379.5 ± 2.83 μg L^−1^, surpassing, by 7.6 times, the MPL set out in NOM-127-SSA1-1994 [5]. The goats in treatment group 3 had drunk the water occurring naturally at their breeding sites throughout their lives. 

#### 2.3.4. Group 4

Additionally, another five goats (conformed group 4) with ages and physiological conditions similar to those in treatment Group 3 were recruited. These control subjects were born in the community of Nuevo Reynosa, a municipality in Matamoros, Coahuila, which has a semi-arid dry climate with, during the season sampled, an average temperature of between 20 to 24 °C. These subjects were selected from a privately-owned mostly female herd, which fed by grazing on vegetation typical of the region, such as candelilla, agave, and native flora. The arsenic concentration in the water was 20.65 ± 0.35 μg L^−1^, which is considered below the MPL stipulated in the regulations. These control subjects were kept in their pen and only drank water naturally occurring in this locality. The exposure period for each treatment lasted 84 days, with samples taken every 21st day to measure the following physiological variables: heart rate; respiratory rate; average body temperature; locomotion; ruminal movement; and, live weight. The effect on the subjects’ libido was determined for treatment Group 1 and compared with its control group (group 2), while the arsenic concentration in the milk produced by the animals of Treatment Group 3 was quantified and compared with the control Group 4.

### 2.4. Arsenic Levels in Urine

The first urine sample was collected on the day of sampling between 5:00 and 6:00 a.m., and, then, in order to ascertain the total arsenic concentration, the urine samples were subject to total acid digestion using the method proposed by Cox, in which digestion is carried out by adding different acids to a heating plate. A calibration curve was generated, with standards of 0.01, 0.025, 0.05, 1.0, and 2.5 μg L^−1^ and using the National Institute of Standards and Technology (NIST) 2670a (High Level Urine 22 μg L^−1^) Standard Reference Material (SRM) as a control. Once the acid digestion was complete, the concentration of arsenic in the urine was analyzed using an atomic fluorescence spectrometer [21].

### 2.5. Physiological Condition

The heart rate (beats/minute), respiratory rate (breaths/minute), average body temperature (anal temperature in °C), locomotion (visual assessment), rumen movement (rumen movement/minute), and live weight (kg) were all measured.

### 2.6. Levels of Arsenic in Milk

The concentration of arsenic in the raw milk produced by the subjects was determined by means of atomic absorption spectrometry (AAS). As, under NOM-117-SSA1-1994 [22], this requires an acid digestion process, the frozen goat′s milk samples were lyophilized using the Labconco^®^ lyophilizer (Kansas City, MO, USA) with a 133 × 10^−3^ mbar vacuum at a temperature of −43 °C. Subsequently, a calibration curve was generated using the Fluka^®^ certified arsenic standard of 1000 μg L^−1^ (Fluka Chemika, Switzerland), at concentrations of 0.05, 0.1, 0.2, 0.25 and 0.3 mg L^−1^. Certified reference materials were used for verifying the calibration and validating the analytical method.

### 2.7. Effect of Arsenic on the Male Goat Libido

A behavioral parameter that determines sexual desire, libido, was determined by placing the male goats in contact with females in heat, taking, as a unit of measurement, the number of mounting attempts (mounts) per treatment group. These measurements were taken during the subjects′ reproductive season, which occurs between the months of September and December.

### 2.8. Statistical Analysis

The statistical analysis was performed via a one-way parametric statistical analysis of variance (ANOVA) performed using SPSS software version 17.0. The level of significance was determined at *p* < 0.01. 

## 3. Results

### 3.1. Concentration of Arsenic in the Water

The concentrations of arsenic in the water varied from 20.6 to 709.3 μg L^−1^, with most of the concentrations observed during the present study above the levels established by the Official Mexican Standard [5]. The maximum value was obtained in the rural community of Benito Juárez (Matamoros, Coahuila), with a concentration of 709.3 ± 64.84 μg L^−1^, which exceeded, by up to 28.37 times, the MPL for arsenic in water established in the Mexican regulations. The rural community of El Venado (Matamoros, Coahuila) presented the second highest arsenic concentration, 379.5 ± 2.83 μg L^−1^, while, in addition to being the study location with the largest population of goats, the rural community of Nuevo Reynosa (Matamoros, Coahuila) presented the lowest concentration, with 20.65 ± 0.35 μg L^−1^. All values exceeded the limit established by both the Environmental Protection Agency of the United States of America and the World Health Organization, whose arsenic MPL is 10 μg L^−1^ [23].

### 3.2. Chronic Toxicity Results for Goat Species Exposed to Arsenic

#### 3.2.1. Results of Group 1 and Group 2

Water: The concentrations of arsenic in the drinking water of the goats in treatment Group 1 were 25.4, 33.35, 31.7 and 27.9 μg L^−1^, on the respective sampling days. 

Urine: After the 84 days that comprised the study, the concentration of arsenic in the urine of the goats administered with sodium arsenite increased significantly (*p* < 0.01) compared to the control, except for the value of 6.21 μg L^−1^ obtained on the last sampling day for treatment group 2 (Table 2). As can be seen, the arsenic concentration in the subjects′ urine is higher in the group administered with a daily dose of arsenic than the concentration recorded for the control group.

Libido: In terms of the response of the libido to arsenic treatment, as represented by mount attempts per animal, no significant differences were found between treatment groups.

#### 3.2.2. Physiological Results for Group 1

The results of Group 1 (Ex situ treatment) show a significant difference (*p* < 0.01) between Group 1 and Group 2 (Table 3). The male goats administered with arsenite presented an average weight reduction of 1.66 kg between the beginning and end of the experimental period, while heart and respiratory frequency increased over the same period. There were no significant differences in temperature or ruminal movement between groups. It should be noted that Treatment Group 1 presented nasal warts and irritability and dehydration in the oral commissure, in contrast with the control group.

#### 3.2.3. Results of Group 3 and Group 4

Water: A high concentration of arsenic was found in the drinking water supply for the goats of the rural community El Venado (Table 4), exceeding the MPL for arsenic concentration in drinking water by several times, while values below the MPL stipulated in the Mexican legislation (NOM-127-SSA1-1994) [5], but exceeding the WHO criteria (WHO 2012) were obtained in Nuevo Reynosa.

Urine: The concentration of arsenic in the urine of goats located in the contaminated area (Group 3) was significantly higher (*p* < 0.01) than the levels found in the control group (Group 4) (Table 4).

#### 3.2.4. Physiological Results of Group 3

The results showed that female Creole goats chronically exposed to high concentrations of arsenic in drinking water developed signs of toxicity, such as weakness and warts around the mouth. The results of the physiological tests showed a significant difference (*p* < 0.01) in weight and respiratory rate between groups. Goats chronically exposed to arsenic presented a lower weight (8%) and higher respiratory frequency (11.78%) than the control group. There were no significant differences (*p* < 0.01) observed in temperature (Table 5).

## 4. Discussion

According to the present study, the Region Lagunera presents high concentrations of arsenic in its water, ranging from 20.6 to 709.3 μg L^−1^, which is well above the MPL stipulated in the Mexican legislation (NOM-127-SSA1-1994), leading to the chronic toxic exposure of its animal population and public health problems. The community with the highest concentration of arsenic was Benito Juárez (709.3 ± 64.8 μg L^−1^), followed by El Venado (379 ± 2.8 μg L^−1^). A water pH of over 7.2 and an electrical conductivity (CE) of 0.36 (mS·cm^−1^) were observed, while the main chemical species found was As V, which is a less toxic form of arsenic. The Region Lagunera is a semi-arid region, where goat farming is an activity of fundamental importance, generally undertaken in areas with poor groundwater quality [24]. It is important to mention that Mexico has the largest goat population in Latin America [25]. Group 1 urine results obtained in the present study indicate the importance of the urinary system in the removal of arsenic from the body, as indicated by previous studies [12,26,27].

Significant differences (*p* < 0.01) were found in the concentration of arsenic between Group 1 and 2. The urine results for Treatment Group 1, which had received an oral dose of sodium arsenite, increased in line with the exposure time (0.6, 3.4, 7.8 and 6.2 μg L^−1^), although the lower level shown by the last result, obtained on Day 84, might indicate an accumulation of arsenic in the blood and tissues, as described in other studies conducted on both goats and rats [12,28]. The female goats exposed to arsenic via their drinking water for more than four years in El Venado (Group 3) presented a higher rate of excretion through urine than the goats subject to Group 1, with an average concentration of 6.89 ± 0.63 μg L^−1^. The concentration of arsenic in the goats′ milk was undetectable in the goats that participated in the present study, from which it can be inferred that this milk and products derived from it would not be harmful to public health. The male goats (Group 1) did not present any effect on libido after exposure to arsenic. While the subjects of Group 3, which had been exposed to drinking water containing high arsenic concentrations, showed a 2.5% decrease in body weight, with this chronic exposure also affecting the respiratory rate and causing heart rate irregularities, as reported by other studies [29]. Treatment groups 1 and 3 also presented behavioral changes, hair loss, and keratosis on the nose and mouth. Some studies refer to the epidemiological significance of arsenic exposure via the water consumed by goats, cows and other species [30,31], highlighting the transmission of this metalloid to humans in high value food products derived from these livestock species, with some of the products described as having high arsenic concentrations including milk, meat, and cheese [32,33,34,35,36]. Moreover, products manufactured from goat and sheep milk, such as yogurt, have be shown to contain metals such as arsenic [37]. In another context, arsenic has been shown to have a genotoxic and immunotoxic effect on goats [38], while effects have also been found on Luteinizing Hormone (LH) and Follicle Stimulating Hormone (FSH) concentrations and sperm quality in Teddy goats exposed to sodium arsenite [39]. In this sense, by means of the consumption of water with high concentrations of arsenic, environmental contamination is a significant problem for animal health and, consequently, for human health via its transmission up the food chain. Our study obtained evidence of these health risks by identifying hyperchromias in the mouths of goats exposed to high concentrations of this metal in their drinking water, as well as observing high levels of arsenic secretion in their urine.

## 5. Conclusions

The Creole goats of the Region Lagunera that have been exposed to chronic arsenic toxicity did not show translocation of the metalloid to their milk, while the male goats did not show libido levels significantly different to the control group. However, the physiological parameters for goats exposed to drinking water with high concentrations of arsenic indicated weight loss, alterations in heart and respiratory rates, behavior changes, hair loss on their flanks, and keratosis on the nose and mouth.

## Figures and Tables

**Table 1 vetsci-07-00059-t001:** Arsenic concentration, pH and Electrical Conductivity (EC) in water samples taken from rural communities in the Region Lagunera.

						Water Taken from the Goat Trough		
Rural Community	Municipality	Geographical Location	Origin of as in Water	pH	CE	As	pH	CE
			(μg·L^−1^)		(mS·cm^−1^)	(μg·L^−1^)		(mS·cm^−1^)
Nuevo Reynosa	Viesca	25°59′0.8″ N, 100°13′17″ W	20.65 ± 0.35	7.63	0.38	23.91 ± 1.30	7.52	0.48
El Cantabro	Francisco I. Madero	25°55′32.9″ N, 103°20′16″ W	33.0675 ± 3.84	7.85	0.39	41.45 ± 0.16	7.78	0.45
Zaragoza	Zaragoza	25°46′31″ N, 103°16′23″ W	36.615 ± 0.021	8.06	0.36	35.49 ± 1.81	7.77	0.38
Finisterre	Francisco I. Madero	25°46′30″ N, 103°21′97″ W	40.17 ± 0.056	7.25	0.62	42.895 ± 4.72	7.26	0.4
Trinidad	Francisco I. Madero	25°46′30″ N, 103°16′28″ W	30.95 ± 1.63	7.22	0.25	31.64 ± 0.85	7.4	1.81
Benito Juárez	Francisco I. Madero	25° 45′ 54.72″ N, 103°36′77″ W	709.3 ± 64.84	7.5	0.38	141.42 ± 2.94	7.5	0.39
Benito Juárez	Matamoros	25° 31′13.44″ N, 103°13′30″ W	22.35 ± 1.48	7.472	0.43	23.1 ± 0.099	8.2	0.49
Corona	Matamoros	25° 31′41″ N, 103°13′42″ W	25.05 ± 2.12	7.58	0.48	11.41 ± 0.55	7.82	0.23
El Consuelo	Matamoros	25° 34′40.8″ N, 103°17′42″ W	23.79 ± 1.07	7.963	0.37	23.185 ± 0.41	8.2	0.47
La Esperanza	Matamoros	25° 33′48.96″ N, 103°17′6″ W	25.6 ± 0.14	7.783	0.45	19.05 ± 0.07	7.49	0.46
Atalaya	Matamoros	25° 39′18″ N, 103°15′18″ W	52.8 ± 14	8.11	0.37	47.225 ± 0.39	8.08	0.41
Solima	Matamoros	25° 42′12″ N, 103°17′24″ W	54.165 ± 0.66	8.056	0.36	51.45 ± 0.35	7.61	0.41
Morelos	Matamoros	25° 30′47.52″ N, 103°12′18″ W	89.65 ± 0.63	7.54	1.17	73.6 ± 0.42	8.1	1.72
El Cariño	Gómez Palacio	25°34′12″ N, 103°28′12″ W	114.05 ± 0.77	8.396	0.28	148.735	7.84	0.24
San Felipe	Gómez Palacio	25°33′48.96″ N, 103° 29.6′ W	32.635 ± 1.96	7.22	0.25	34.83	8.3	0.24
Charcos de Risa	Francisco I. Madero	26°12′54.72″ N, 103°06′18″ W	142.655 ± 1.92	9.311	2.7	142.1	9.21	2.7
El Venado	Francisco I. Madero	26°50′ N, 101°58′ W	379.5 ± 2.83	8.15	1.26	262.815	7.3	1.07

**Table 2 vetsci-07-00059-t002:** Effects of arsenic on male Creole goats of the Region Lagunera– Ex situ treatment.

	Group 1: As Treatment				Group 2: Control			
	Day 1	Day 28	Day 56	Day 84	Day 1	Day 28	Day 56	Day 84
Water (μg L^−1^)	25.4 ± 0.07	23.7 ± 0.1	31.3 ± 1.2 *	22.07 ± 1.2	25.4 ± 0.03	23.35 ± 0.08	31.3 ± 1.2 *	21.07 ± 0.9
Urine (μg L^−1^)	0.6 ± 0.03	3.4 ± 0.05	7.8 ± 1.1 *	6.21 ± 0.9 *	0.02 ± 0.002	0.029 ± 0.01	0.042 ± 0.035 *	0.06 ± 0.02 *
Libido (Mounting attempts)	5	5	5	4	5	5	5	4

Mean ± SE; * *p* < 0.01.

**Table 3 vetsci-07-00059-t003:** Effects of arsenic on body weight and physiological conditions in Group 1 vs. Group 2.

	Group 1: As Treatment				Group 2: Control			
	Day 1	Day 28	Day 56	Day 84	Day 1	Day 28	Day 56	Day 84
Body weight (kg)	63.74 ± 4.8	62.12 ± 3.2	62.08 ± 3.8 *	62.08 ± 3.9 *	64.02 ± 2.6	63.5 ± 2.4	64.96 ± 3.1	65.42 ± 3.6 *
Heart frequency	66.8 ± 1.3	72.8 ± 10.2 *	78.8 ± 2.1 *	78.2 ± 1.7 *	69.6 ± 0.6	69.4 ± 1.2 *	67.4 ± 1.7 *	68.6 ± 1.0 *
Respiratory frequency	25.5 ± 0.5	27.6 ± 0.8 *	28.8 ± 1.4 *	31.2 ± 1.1 *	21.68 ± 1.8	20.8 ± 2.2 *	19.8 ± 3.0 *	23.8 ± 1.7 *
Temperature	38.5 ± 0.0	38.5 ± 0.5	38.7 ± 0.2 *	38.7 ± 0.2 *	38.5 ± 0.02	38.5 ± 0.03	38.6 ± 0.07	38.5 ± 0.5
Ruminal movement	1.71 ± 0.2	1.8 ± 0.3	1.86 ± 0.4	1.85 ± 0.3	1.8 ± 0.8	1.8 ± 0.5	1.75 ± 0.5	1.86 ± 0.3

Mean ± SE; * *p* < 0.01.

**Table 4 vetsci-07-00059-t004:** Effects of arsenic on female Creole goats of Groups 3 vs. 4.

	Group 3: Contaminated Area–El Venado				Group 4: Control Area–Nuevo Reynosa			
Days	1	28	56	84	1	28	56	84
Water (μg L^−1^)	262.8 ± 14	441.8 ± 7.6 *	377.5 ± 0.7 *	381.5 ± 2.8 *	20.7 ± 0.04	25.8 ± 1.07 *	19.4 ± 0.003	21.2 ± 0.06
Urine (μg L^−1^)	6.6 ± 0.02	7.81 ± 0.16 *	6.78 ± 0.3	6.4 ± 0.002	0.15 ± 0.00	0.13 ± 0.02	0.16 ± 0.01	0.16 ± 0.02
Milk (μg L^−1^)	N. D.	N. D.	N. D.	N. D.	N. D.	N. D.	N. D.	N. D.

Mean ± SE; * *p* < 0.01; N. D. (Not Detectable).

**Table 5 vetsci-07-00059-t005:** Effects of arsenic on physiological conditions in female Creole goats in a contaminated zone and control area (Groups 3 vs. 4).

	Group 3: Contaminated Area–El Venado				Group 4: Control Area–Nuevo Reynosa			
Days	1	28	56	84	1	28	56	84
Body weight (kg)	43.2 ± 3.9	43.5 ± 4.1	42.2 ± 5.2 *	41.6 ± 4.3 *	44.92 ± 1.5	42 ± 1.7	47.1 ± 2.1 *	48.95 ± 1.7 *
Heart frequency	68.6 ± 1.0	64.2 ± 1.5 *	62.4 ± 2.0 *	60.6 ± 1.6 *	65.4 ± 2.3	63.7 ± 1.4 *	62.9 ± 1.2 *	63.2 ± 1.2 *
Respiratory frequency	31.5 ± 3.0	37.2 ± 1.5 *	38.3 ± 2.0 *	33.1 ± 2.6 *	29.7 ± 3.0	31.6 ± 2.8 *	32.9 ± 1.8 *	29.5 ± 1.5
Temperature	38.5 ± 0.05	38.6 ± 0.03	38.6 ± 0.3	38.5 ± 0.07	38.5 ± 0.06	38.5 ± 0.03	38.4 ± 0.03	38.5 ± 0.02
Ruminal movement	1.78 ± 0.03	1.8 ± 0.09 *	1.86 ± 1.1 *	1.85 ± 1.2 *	1.76 ± 0.03	1.86 ± 0.04	1.86 ± 0.05	1.85 ± 0.07

Mean ± SE; * *p* < 0.01.

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
