# Peer review of "Toxicity in Goats Exposed to Arsenic in the Region Lagunera, Northern Mexico"

_vetsci, 2020, doi:10.3390/vetsci7020059_

Round 1

Reviewer 1 Report

The paper was adequately improved. At proofing please check the superscripts in units (i.e. -1). The attribute is often missing

Author Response

Reviewer 1

We appreciate your comments and improvements for this manuscript.

All new changes are highlighted in purple.

All superscripts units have been corrected.

Reviewer 2 Report

  • Abstract should be corrected, because still it is not clear how many animals were used for this study. 
  • The description of the each group of animals should be unified in the whole manuscript. For example: 1. Treatment 1 (E.S.) - group I and group II; 2. Treatment 2 (I.S.)- group III and group IV 
  • How many animals were in each group? because there is an information  that the experiment was performed on the 20 animals in the Treatment 1 (E.S.) group. 
  • Table 3 and Table 4 - please check once again the description of the each group of animals: there is a lack of the group III and group IV in the description of the materials and methods 
  • The resolution of Figure S1 is still low.

Author Response

Reviewer 2

We appreciate your comments and improvements for this manuscript.

All new changes are highlighted in purple

1.- The summary has clarified the number of animals used in the study, as well as in each group.

2.- The description of each group of animals has been unified as indicated. The groups have been as: Group 1, group 2, group 3 and group 4.

3.- For each group, the number of animals that were used per group has been indicated.

4.- For Table 2, Table 3, Table 4 and Table 5 they have been corrected according to the allocation nomenclature for each group.

5.- The description of groups 3 and 4 has been included in the materials and methods section.

6.- The resolution of Figure S1 has been improved to 300 dpi.

7.- In the discussion section, corrections for study groups have been included.

We appreciate comments and questions to the manuscript, hoping they have been answered appropriately.

This manuscript is a resubmission of an earlier submission. The following is a list of the peer review reports and author responses from that submission.

Round 1

Reviewer 1 Report

The paper is relevant to the journal addressing an important issue of endemic exposure to As. At the same time, the quality of the paper's preparation is quite poor and interferes with clarity and scientific soundness.

Authors mix up experiments on male and female goats together. In the abstract it is stated that the goats were males but then milk was analysed. In the main body it turns up that in in situ experiments also females were involved. Table 1 is a draft, it contains some stuff of preliminary preparation 'Coordenada?'. Most of the paper is like that. The language and style is also not adequate for publishing and require improvement.

Specific comments

1) Validation data for As quantification (ref. material analysis) should be included in the paper (e.g. as supplement).

2) Ethical approval is not disclosed.

3) Introduction: As endemic exposure is important but unlikely 'most important health problems worldwide'. What is 'like damage to body and physiological condition'? It's too general, please, be more specific. What is 'loss of condition' few lines later? The part of the paragraph on Creol goats is not connected to the rest of the paragraph.

4) Section 2.3.2. What is 'wild flora'? Revise "herd was property individuals dedicated to goat production".

At this point, it is hard to evaluate the discussion at this point due to the lack of clarity in the results. The paper requires major revision.

Reviewer 2 Report

How is it possible to write a scientific article in this form? No comment!

Reviewer 3 Report

The manuscript entitled: „Toxicity in goats exposed to Arsenic in the Region Lagunera, Northern of Mexico” is interesting and with new data, however in my opinion this manuscript needs revision.

Affiliation of authors:

“*Corresponding author: Dr. Javier Morán martínez…”- should be improved as: „*Corresponding author: Dr. Javier Morán Martínez…”

Abstract:

“Sodium arsenite (2 mg / Kg)…” - please use the unit: mg/kg, check this in the whole manuscript and unified (for example: “0.2 mg Kg-1” should be change as “0.2 mg kg-1”).

“we found significative differences…”- please change as: “We found significative differences…”

Introduction:

“…while World Health Organization (WHO) indicates 10 μg L-1 ([6].”- please change as: “…while World Health Organization (WHO) indicates 10 μg L-1 [6].”

Material and methods:

“…with a total of 48 887.50 Km2 (13).” - please change as: “…with a total of 48 887.50 km2 (13).”

Table 1- I couldn`t check this table because it was not possible to see the whole text in this table. Please attached this table once again.

“…with a weight of 60-70 Kg each.” - please change the unit as: kg.

“…municipality of Matamoros, Coahuila (rural community with the highest concentration of arsenic and high number of goats), These were kept in their original pens.” - please change as: „…municipality of Matamoros, Coahuila (rural community with the highest concentration of arsenic and high number of goats). These were kept in their original pens.”

I couldn`t find in this section the permission of the Ethics Committee for these experiments. Please add the number of this permission.

Results:

In the description of the results please use: Table 1 etc. not table 1.

References: According to authors guideline the References should be unified. Please check.

I couldn`t check the Supplementary Materials: “Figure S1: Map of concentration of arsenic in the water of the Comarca Lagunera. Video S1: Goat with white muscle disease”.
